# The level of protein in the maternal murine diet modulates the facial appearance of the offspring via mTORC1 signaling

The development of craniofacial skeletal structures is fascinatingly complex and elucidation of the underlying mechanisms will not only provide novel scientific insights, but also help develop more effective clinical approaches to the treatment and/or prevention of the numerous congenital craniofacial malformations. To this end, we performed a genome-wide analysis of RNA transcription from non-coding regulatory elements by CAGE-sequencing of the facial mesenchyme of human embryos and cross-checked the active enhancers thus identified against genes, identified by GWAS for the normal range human facial appearance. Among the identified active cis-enhancers, several belonged to the components of the PI3/AKT/mTORC1/autophagy pathway. To assess the functional role of this pathway, we manipulated it both genetically and pharmacologically in mice and zebrafish. These experiments revealed that mTORC1 signaling modulates craniofacial shaping at the stage of skeletal mesenchymal condensations, with subsequent fine-tuning during clonal intercalation. This ability of mTORC1 pathway to modulate facial shaping, along with its evolutionary conservation and ability to sense external stimuli, in particular dietary amino acids, indicate that the mTORC1 pathway may play a role in facial phenotypic plasticity. Indeed, the level of protein in the diet of pregnant female mice influenced the activity of mTORC1 in fetal craniofacial structures and altered the size of skeletogenic clones, thus exerting an impact on the local geometry and craniofacial shaping. Overall, our findings indicate that the mTORC1 signaling pathway is involved in the effect of environmental conditions on the shaping of craniofacial structures.

A key aspect of most social communication between humans is facial recognition[1]. Accordingly, congenital craniofacial malformations, including cleft palate, craniosynostosis, and craniofacial skeletal hypoplasia, which together account for more than one-third of all congenital birth defects, can have a profound influence on social interactions[2,3]. From an evolutionary perspective, the viscerocranium harbors vital structures such as the feeding apparatus and supports sensory organs. Its precise sculpting and inheritable reproducibility are of unequivocal importance for survival. At the same time,

adaptability of viscerocranium and particularly of the feeding apparatus to the environmental cues allows adjusting to environmental changes, e.g., feeding sources, or even concurring novel ecological niches. The latter type of adaptation is less common in mammals than in several classes of Gnathostomata, e.g., Actinopterygii[4].

The craniofacial skeleton, one of the most complex and sophisticated part of the skeletal system, is composed of many different parts and formed via the interplay of a variety of genetic, epigenetic, and environmental factors[5–7]. Studies involving families with monozygotic

✉ e-mail: igor.adameyko@meduniwien.ac.at; andrei.chagin@gu.se

and dizygotic twins indicate that the genetic inheritability score for craniofacial morphology in humans varies widely among different facial features, e.g., from 0.8 for the distance between the inner corners of the eyes to approximately 0.5 for the position of the point midway between the nose and upper lip, as well as for nasal protrusion[8–11]. One well-known environmental influence on facial morphogenesis in humans is alcohol consumption during pregnancy[12].

In all *Gnathostomata*, including zebrafish, mice, and humans, the viscerocranium develops from descendants of the neural crest cells (NCCs), a transient and multipotent population of embryonic progenitors[13]. Multiple subpopulations of the mesenchyme derived from NCCs condense and then differentiate further into chondrocytes and osteoblasts, the major types of skeletal cells in the craniofacial region[14–16]. Other developmental subpopulations derived from NCCs, such as Schwann cell precursors, also contribute to the formation of chondrocytes and osteoblasts, but to a relatively limited extent[17]. The shapes of the craniofacial skeletal elements are determined primarily by the shape of these mesenchymal chondrogenic condensations during development, with subsequent fine-tuning by localized intercalation of new chondrogenic clones arising from the surrounding mesenchyme[18,19].

A complex interplay between NC cells, the facial ectoderm, placodes, endoderm and neuroepithelium orchestrates accurate sculpturing of the viscerocranium. Not surprisingly, this process involves continuous changes in the expression of thousands of different genes[20]. Genome-wide association studies (GWAS) have implicated more than 100 loci in the formation of facial morphology within the normal range and more than two hundred single nucleotide polymorphisms (SNPs) that exert a significant impact on this formation[21–23]. In association with abnormal morphology of the human facial skeleton, the HPO database (http://hpo.jax.org) lists 1165 genes[24]. Although many of these are associated primarily with other systems, such as hematopoiesis and neurogenesis, their large number reflects the complexity of facial morphogenesis.

At the same time, the proper migration and differentiation of NCCs, as well as their interaction with surrounding tissues during facial development involves limited number of developmentally and evolutionarily conserved signaling pathways. Among those are hedgehog (HH), fibroblast growth factors (FGF), bone morphogenic proteins (BMPs), WNT, retinoic acid (RA) and platelet-derived growth factor (PDGF) pathways[25]. These pathways and associated morphogens form a hardware system, genetically responsible for sculpting of viscerocranium. However, the signaling pathways, capable to sense environmental clues and integrate these signals into genetical hardware of facial morphogenesis are rather unknown. One proposed system is HH signaling pathway, which may sense mechanical forces via cilium[26] and modify craniofacial formation as it has been shown in bony fish[27].

Clearly, a major factor underlying natural selection has been the availability of nutrition[28], and the feeding apparatus, part of the viscerocranium, is of particular importance in this context. Nutritional sensing by the Mechanistic Target of Rapamycin Complex 1 (mTORC1) signaling pathway has been highly conserved evolutionarily[29]. Budding yeasts sense the availability of amino acids via mTORC1 and, in response to this information, shift towards the synthesis of proteins or autophagy[29]. Although this pathway plays a similar role in multicellular organisms[30,31], in this case, levels of oxygen, energy, and growth factors (primarily those transducing via PI3 kinase and AKT[30]) also exert an influence. Indeed, in mammals, mTORC1 is activated in vivo by protein intake and this activation correlates most with the serum levels of both branched-chain amino acids and glucose[32] reflecting multiple levels of controlling mTORC1 activity both intra- and extracellularly, such as cell energy sensor AMP-activated protein kinase (AMPK) and growth factors[33]. Among the latter, insulin growth factors (IGFs) are major regulators of mTORC1 pathway, activating it via the PI3/Akt pathway[31].

On the systemic level, mTORC1 system can be regarded as an amino-acid sensing machinery belonging to an endocrine network of both up- and downstream of IGFs that regulates a variety of processes in response to the availability of nutrition, including intracellular protein synthesis and autophagy[30,34–36]. Changes in the activity of mTORC1 can alter the shape of craniofacial structures[37,38] and, in addition, the mTORC1 pathway interacts with the HH, BMPs, and Wnts[39–41] pathways, which are strongly involved in sculpting the viscerocranium. Accordingly, we hypothesize that the mTORC1 signaling pathway may play a role in mediating interactions between certain environmental factors and the inherited program of craniofacial morphogenesis.

## Results

### PI3K/mTORC1 pathway is linked to facial appearance in humans

To identify transcribed enhancers actively involved in human facial development, embryonal facial material was CAGE-sequenced (Fig. 1A) and all enhancers actively transcribed during human facial development between weeks 3 and 12 of gestation were identified (supplementary metadata files can be found here https://zenodo.org/records/10363659). All thus identified enhancers were further cross-checked and enriched against enhancers previously identified in ENCODE project[42] (see the Methods). The resulting pool of enhancer coordinates was overlapped with published GWAS hits[22] (see the Methods) identified to be associated with normal-range facial morphology (Fig. 1A). Among enhancers thus identified there was a clear enrichment in components of the PI3K/AKT/mTORC1/autophagy pathway (Fig. 1B, C, Supplementary Fig. 1). Predictions based on the STRING database (Fig. 1D) and the specific facial phenotype related to each individual polymorphism (Fig. 1E) indicated that among the major enhancers active during sculpting of the human face, the PI3K/AKT/mTORC1 pathway was clearly enriched.

Thus, this approach identified the PI3K/AKT/mTORC1/autophagy pathway as a potentially important player in human facial morphogenesis. To explore the mechanism(s) underlying the involvement of this pathway in craniofacial shaping, we manipulated the mTORC1 pathway during facial development in experimental animals.

### mTORC1 modulates the shaping of chondrogenic condensation in mice

First, we activated the mTORC1 pathway in neural crest cells (NCCs) by crossing *Tsc1* floxed mice with the *Sox10^CreERT2* strain[43], in which a pulse of tamoxifen on embryonic day 8.5 (E8.5) causes recombination in NCCs[15]. Reconstruction of the developing craniofacial structures in the offspring utilizing 3D μ-CT images with enhanced contrasting of soft tissues revealed alterations in the thickness of skeletal elements, as well as minor developmental abnormalities already on E17.5 (Fig. 2A–D). Overlay of the reconstructed cartilage of *Tsc1* cKO and control (*Tsc1* heterozygous) embryos revealed enlargement of a variety of elements of the craniofacial skeleton, as well as enhanced thickness of all components of the nasal cartilage (Fig. 2E, F). These observations confirmed the involvement of the mTORC1 pathway in craniofacial shaping[37,38] and, in addition, showed that this pathway is involved during early development.

Previously, we demonstrated that in mice craniofacial shape is established at the time of mesenchymal condensation (E12.5-E13.5), with subsequent fine-tuning via intercalation of new clones[18,19]. To reveal the shape of these mesenchymal condensations, KO embryos were stained for Sox9 on E12.5 and, although the overall shape was preserved, their nasal prominence and nasal capsule compartments were thicker (Fig. 2G–I). We then bred in the *R26R^Confetti* reporter transgene, which allows clonal behavior to be assessed. Analysis of *Sox10^CreERT2;Tsc1^flfl;R26R^Confetti* embryos on E17.5 (with previous pulsing on E8.5) revealed that in the absence of the *Tsc1* gene, the clones of nasal chondrocytes appeared as bulky large clusters (Fig. 2J–O), with

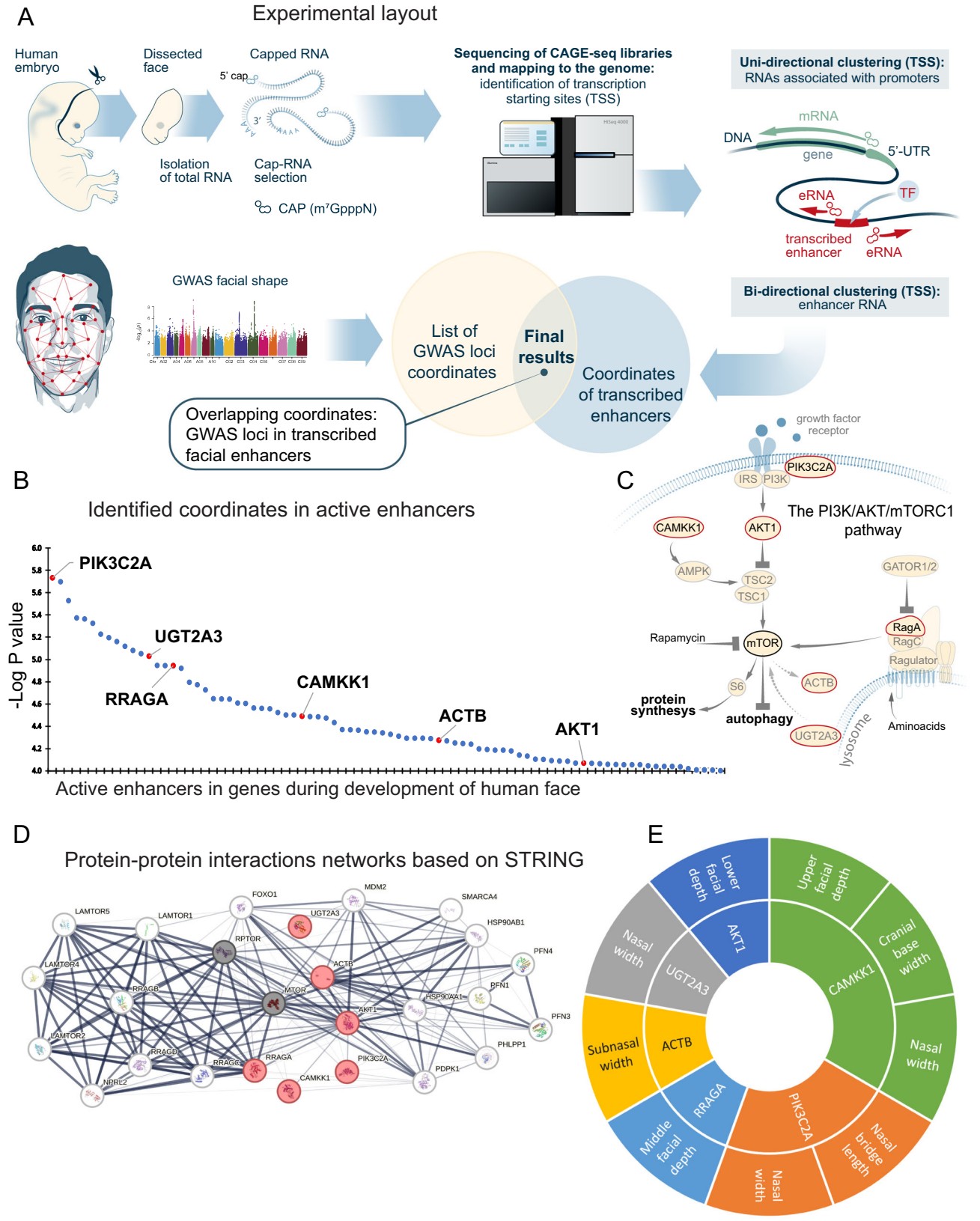

**A** Experimental layout

**B** Identified coordinates in active enhancers

**C** The PI3K/AKT/mTORC1 pathway

**D** Protein-protein interactions networks based on STRING

**E**

more extensive dispersion and misalignment (Fig. 2P–S), compared to the individual columns observed in the *Sox10^{CreERT2};Tsc1^{fl+};R26R^{Confetti}* heterozygotes (Fig. 2J–S).

Thus, activation of the mTORC1 pathway in murine NCCs modulated both chondrogenic condensation and clonal arrangement.

**mTORC1 minorly affects facial shaping at intercalation stage**

To explore the influence of the mTORC1 pathway during intercalation of new clones into existing mesenchymal condensations[19], we injected tamoxifen on E12.5, the stage at which *Sox10^{CreERT2}* targets perichondrial cells surrounding cartilage elements and Schwann cell precursors[17].

**Fig. 1 | Active enhancers in human embryonic tissue associated with facial individuality. A** Experimental layout: embryonic human facial mesenchyme was collected and CAGE-sequenced to identify transcribed enhancers, which were subsequently compared to SNPs associated with facial morphology within the normal range as indicated by genome-wide association sequencing (GWAS). **B** Distribution of the genes located proximally to the enhancers identified in relationship to the GWAS probability. Genes related to the PI3K/AKT/mTORC1/autophagy pathway are highlighted in red. **C** Illustration of the PI3K/AKT/mTORC1/ autophagy pathway (based on refs. 29 and 17 and the Autophagy database http://www.tanpaku.org/autophagy/) highlighting the genes identified. **D** Protein interaction network generated by the STRING database for the genes related to the development of individual facial features. The six genes indicated in B are highlighted in red. The thickness of the connecting lines reflects the strength of interactions. **E** Distribution of the individual facial phenotypes revealed by GWAS to be associated with each of the six genes encoding components of the PI3K/AKT/mTORC1/autophagy pathway.

Surprisingly, ablation of Tsc1 at this developmental stage only increased clonal size slightly (Fig. 3A–J). To further verify this observation, we targeted chondro-progenitors involved in early mesenchymal condensation employing $Col2^{CreERT}$ mice coupled with both the $Tsc1$ floxed and $R26R^{Confetti}$ strains and pulsed with tamoxifen on E12.5. In line with the previous observation, activation of mTORC1 signaling by $Col2^{CreERT}$ at this developmental stage did not alter the structure of the craniofacial skeleton and increased clonal size slightly (Fig. 3K–O).

At the same time, ablation of mTORC1 signaling in chondro-progenitors by crossing $Raptor$ floxed mice with the $Col2^{CreERT}$ and $R26R^{Confetti}$ strains and pulsing with tamoxifen on E12.5 augmented facial length on E17.5 (as detected by μCT, see Fig. 3P–R) without affecting any other skeletal parameters (Supplementary Fig. 2A–C). Ablation of Raptor under these same conditions lowered the number of large clones somewhat and enhanced the number of cells that were single-labeled (Fig. 3S–Y). Successful manipulation of mTORC1 activity in these various strains was confirmed by assessment of S6 phosphorylation (Supplementary Fig. 2D–E).

These observations indicate that in mice, the mTORC1 pathway is involved in craniofacial shaping predominantly prior to and/or during the stage at which chondrogenic condensations occur.

### mTORC1 impacts pre-condensation stage in mice and zebrafish

To establish the role of mTORC1 signaling at the stage prior to chondrogenic condensation, we inhibited mTORC1 with a single injection of rapamycin into pregnant animals on E10.5, when migration of cranial NCCs has been completed, but chondrogenic condensation has not yet begun[15,17]. This resulted in a slightly elongated snout in the embryos on E17.5 in comparison to the controls injected with DMSO (Fig. 4A–C). Moreover, the thickness of chondrogenic mesenchymal condensations on E12.5 was reduced (Fig. 4D–F). Clonal lineage tracing of chondro-progenitors in these same embryos beginning on E12.5 revealed disorganized clones, with relatively fewer elongated clones containing more than three chondrocytes and a relatively higher number of labeled cells that had not divided (i.e., in which recombination had occurred, but which did not proliferate during the period of tracing) (Fig. 4G–M).

To determine whether this influence of mTORC1 signaling on mesenchymal condensation is conserved among species, we also studied zebrafish, in which shaping of the craniofacial skeleton also occurs via chondrogenic condensation and intercalation of chondroprogenitors into the primary cartilage anlagen[44]. In these animals the craniofacial skeleton begins to develop between 48 and 72 hours postfertilization (hpf)[45] and the first $Sox9$- and $Col2$-positive cells appear at 48 and 53 hpf, respectively[17]. $Col2a1aBAC:mcherry$ zebrafish larvae were exposed to rapamycin at various time-points in their development, washed free of this compound, and then allowed to develop until 120 hpf. Exposure prior to (14–22 hpf, 24–32 hpf) or during (32-56 hpf) chondrogenic condensation did not affect the overall size of the facial skeleton, but led to narrowing of cartilaginous structures (Fig. 4N–S). Interestingly, significant elongation of the face occurred when the larvae were exposed to rapamycin at 14–22 hpf or 32–48 hpf (Fig. 4R). Furthermore, exposure prior to chondrogenic condensation resulted in slight curvature of the ethmoid plate (ETH) and repositioning of several other elements of the cartilage (MC, PQ and CH) (Fig. 4P, Q).

Altogether, these findings indicate that in both mice and zebrafish the mTORC1 pathway modulates the shape of craniofacial structures by regulating the recruitment and clonal expansion of mesenchymal derivatives of neural crest cells. Interestingly, even transient inhibition of mTORC1 activity early during development altered the clonal behavior of NCC progeny, thereby leading to subsequent modulation of the shape of the craniofacial skeleton.

### Dietary interventions alter facial structure via mTORC1

The evolutionary conservation and mild variability described above indicate that mTORC1-dependent modulation of craniofacial structures, and particularly those of the feeding apparatus, may be an important adaptive mechanism. As also mentioned above, the activity of the mTORC1 pathway is regulated by nutritional status and, in particular, by dietary levels of amino acids, which act both directly at the cellular level through receptors for arginine and leucine and systemically via pathways involving growth hormone and insulin growth factors (IGFs), which are themselves also controlled by amino acids levels[30,34]. Accordingly, we examined whether alteration of mTORC1 activity through feeding diets containing different levels of protein to pregnant dams might modulate craniofacial shaping in the offspring. For this purpose, starting on E6.5 pregnant C57BL/6 J mice consumed isocaloric diets containing either 20% protein (a level similar to that in standard mouse chow = the control), 4% (low) protein, or 40% (high) protein, with subsequent analysis of at least 4 different litters of embryos from each group.

As expected, mTORC1 activity (as reflected in the level of pS6) was lowest in the embryos whose mothers were subjected to low-protein diet and most pronounced in the high group (Supplementary Fig. 3A, B), with no differences in body weight (Supplementary Fig. 3C). μCT scans of embryos on E17.5 utilizing phosphotungstic acid (PTA) to augment contrast (Fig. 5A) revealed that both the length and width of the nasal capsule (Fig. 5B, C), as well as the length of the Meckel's cartilage (Fig. 5D) were all influenced by the level of protein in the maternal diet. Thus, comparison of 3D segments of the chondrocranium cartilage showed that both the nasal capsule and mandible were slightly smaller in the embryos whose dams received 4% dietary protein (Fig. 5E). In addition, the thickness of the cartilage of the nasal capsule was elevated by the higher level of dietary protein (Fig. 5F). To confirm these observations, the same experiment was performed, but utilizing Hexabrix 320 for contrast in connection with the μCT, and similar changes in craniofacial structures were observed (Supplementary Fig. 3D–M). Lowered proliferation of cells within skeletal elements was observed only in embryos whose dams received the lowest level of dietary protein (Supplementary Fig. 3N–T) and there were no differences between the groups with respect to the extent of cell death within cartilaginous elements (Supplementary Fig. 3Q-U).

When the level of protein in the diets was manipulated in this same manner in pregnant $Sox10^{CreERT2};Tsc1^{fl/fl}$ dams (pulsed with tamoxifen on E8.5 and, accordingly, having embryos with constitutively active mTORC1 in all their NCCs-derived cells) and the craniofacial structures of these embryos analyzed on E17.5, again by μCT scans utilizing PTA for contrasting, no differences in any of craniofacial parameters were detected (Fig. 5G–J). Thus, the lack of alteration in the craniofacial parameters examined (Fig. 5H–J), together with the changes in pS6 activity observed above (Supplementary Fig. 3A, B),

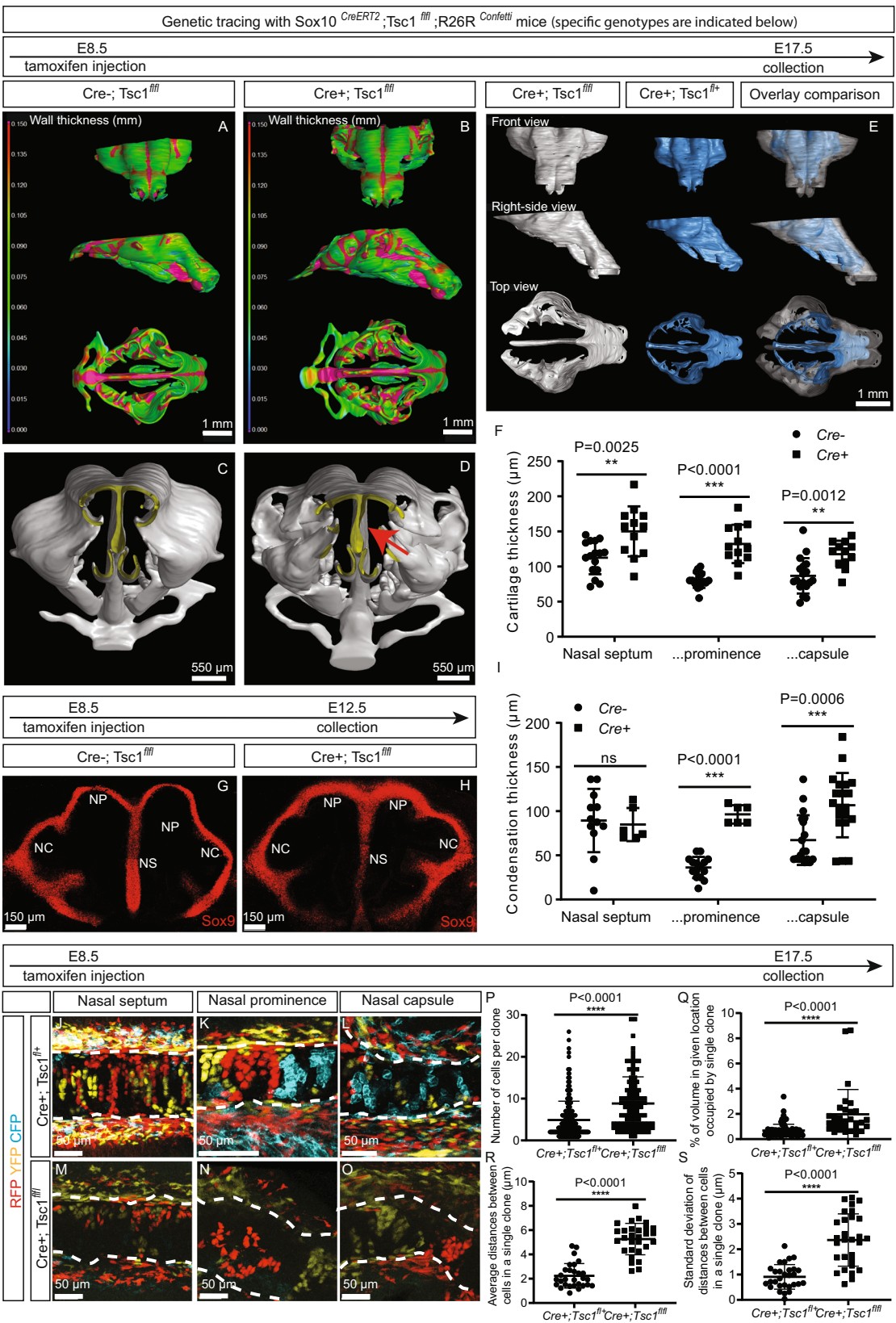

suggests that the alterations in craniofacial structures in response to the different levels of dietary protein were mediated by mTORC1 signaling.

Next, when the level of protein was manipulated in this same manner in the diets of *Col2^{CreERT2};R26R^{Confetti}* mice and these animals injected with tamoxifen on E12.5 and E13.5, both the low and high levels

of protein caused remarkable disorganization of the clones within developing cartilaginous elements (Fig. 5K–L). This finding agrees with the conclusion above that both elevation and attenuation of mTORC1 activity disturb clonal organization within developing cartilage.

Finally, the incorporation of the average values obtained in mice with the low and high protein diets to a mathematical model of human

**Fig. 2 | Activation of the mTORC1 pathway in the neural crest cells of mice results in thickening of the nasal cartilage, mesenchymal condensations, and the formation of bulky clonal clusters. A–D** The mTORC1 pathway in *Sox10^{CreERT2};Tsc1^{fl/fl};R26R^{Confetti}* mice was activated by ablation of *Tsc1* through exposure to tamoxifen on E8.5. 3D reconstruction of the entire chondrocranium on the basis of μCT with enhancement by PTA allowed comparison of the thickness of various structures in control **A, C** and *Tsc1cKO* **B, D** mice. The surface of the skeletal structures as revealed by these reconstructions in control and *Tsc1cKO* mice are shown in **C, D**. The arrow in **D** points toward the curved nasal septum. **E** Overlay of 3D-reconstructions revealed differences in the size and shape of the chondrocranium in control and Tsc1cKO mice. **F** Quantification of thickness of the cartilage in the three major nasal compartments. *n* = 12 animals.
**G–H** Representative images of chondrogenic condensations (revealed by SOX9

staining on E12.5) in control **G** and Tsc1 cKO **H** mice. **I** Quantification of the condensation thickness of the three major nasal compartments. *n* = 12 animals.
**J–O** Clonal arrangements in facial cartilage on E17.5 in the presence of one (J-L, control) or no **M–O** copy of the *Tsc1 gene* (*Tsc1cKO*) are depicted. The images shown are representative for the three major types of nasal cartilage—the septum **J, M**, prominence **K, N**, and capsule **L, O**. **P–S** Clones within cartilage were characterized with respect to the number of cells per clone **P**, volume occupied by each clone **Q**, average distance between cells within each clone **R**, and dispersion of the clones as reflected in the standard deviation of distances between cells **S**. Between 30 and 249 clones obtained from 3 animals were quantified **P–S**. Means ± SD are presented for all quantification graphs. Two-sided student's unpaired t-test was applied to the values in **F, I, P, Q, R, S**. Source data are provided as a Source Data file.

skulls for visualization purposes indicated slight but clear alterations in multiple elements of the craniofacial skeletons (Supplementary Fig. 4, see the Methods for further details).

Altogether, these findings indicate that the level of protein in the maternal murine diet during pregnancy influences embryonic shaping of craniofacial cartilage, which is likely mediated by altering the activity of mTORC1, which in turn changes the clonal dynamics of neural crest progeny.

## Discussion

Here, we have revealed cellular mechanisms underlying mTORC1-dependent shaping of elements of the craniofacial skeleton and demonstrated in both zebrafish and mice that this shaping occurs predominantly in association with mesenchymal chondrogenic condensations, with subsequent fine-tuning to a lesser degree via intercalation. In addition, we have demonstrated that mTORC1 activity in mouse embryos is modulated by the level of protein in the maternal diet, with associated effects on the chondrocranium and fine-tuning of the shape of the craniofacial skeleton.

In greater detail, we show here that alterations in the behavior of progeny of NCCs influence skeletal shaping, both at the stage when chondrogenic mesenchymal condensations occur and when the 3D morphology of the cranial skeleton is fine-tuned via clonal intercalation. The finding that the shape of mesenchymal condensations largely determines the subsequent shape of cartilaginous and, later, bony structures[19] provides a link between the expansion of ectomesenchyme derived from murine NCCs lacking *Tsc1* and resulting changes in craniofacial shape[35]. It is noteworthy that mTORC1 activity influences the shaping of different chondrogenic mesenchymal condensations to different extents, with imperfect preservation of the rough 3D geometry of the entire chondrocranium. For example, constitutively active mTORC1 increases the thickness of the condensations underlying the nasal prominence and nasal capsule, while changing the patterning of the nasal septum to a much more limited degree.

Therefore, specific mechanisms or processes appear to be localized within distinct regions of the chondrocranium. The potential underlying mechanism(s) may involve the known interactions between the mTORC1 pathway and the major morphogens involved in the shaping of craniofacial structures, including HHs (hedgehogs), FGFs (fibroblast growth factors), BMPs (bone morphogenetic proteins), WNTs (Wingless/Integrated family of morphogens), RA (retinoic acid) and PDGFs (platelet-derived growth factors)[25,33,41,46–49]. For instance, ablation of mTOR specifically in NCCs reduces the activities of the canonical Wnt and BMP pathways[38]. SHH, which is secreted in localized regions by the neuroepithelium and the brain, participates in shaping the anterior chondrocranium in a highly specific manner, e.g., by inducing or permitting formation of the nasal septum[18]. At the same time, S6K1, a kinase downstream of mTORC, augments HH signaling by phosphorylating GLI1[41]. Thus, the differential chondrogenic activity of SHH, in combination with its functional interactions with the mTORC1 pathway, may contribute to the difference in the consequences of

chondrogenic condensations at different locations in the developing face.

Furthermore, our present findings indicate that the mTORC1 pathway influences facial skeletal shaping by modulating the clonal expansion of committed chondro-progenitors. Previously, we reported that the growth of facial skeletal elements depends on the intercalation of chondrocyte clones originating from committed chondro-progenitors within the perichondrium surrounding these elements and oriented transversally into pre-formed cartilage[19]. This intercalation and subsequent expansion of chondrogenic clones plays a key role in controlling the final thickness and geometry of cartilaginous elements. Here, we show that manipulation of mTORC1 activity prior to the formation of the perichondrium and committed chondro-progenitors alters the formation of these oriented clones later in development. With attenuated mTORC1 activity, the intercalated clones in the nasal cartilage of embryos were smaller, whereas elevation of mTORC1 activity in chondro-progenitors via deletion of the *Tsc1* gene resulted in intercalation of bulky clonal clusters rather than individual clonal columns. Intercalation of these aberrant clones likely underlines the altered length and thickness of the nasal cartilage, which eventually influenced the overall craniofacial shape. Thus, during mesenchymal condensation mTORC1 activity regulates the overall geometry of facial cartilage, whereas with respect to committed chondro-progenitors this activity influences individual cartilaginous elements. Therefore, modulation of mTORC1 activity at different time-points may result in a spectrum of somewhat different craniofacial shapes, perhaps thereby also contributing to the variety of defects in patterning observed.

It is worth pointing out that the mTORC1 pathway is also involved in chondrogenesis in the limbs, with ablation of Raptor in the limb bud mesenchyme resulting in growth impairment[50]. However, modulation of mTORC1 activity in mature chondrocytes does not influence limb growth[51,52]. These observations indicate that the appropriate level of mTORC1 activity in chondro-progenitors, rather than in mature chondrocytes, is important for skeletogenesis, in line with our present results.

Since mTORC1 is primarily involved in adjusting cellular responses to the nutrition available[53,54], either being enhanced directly by amino acids[32,55] or via insulin and insulin-like growth factors (all of which are tightly regulated by nutritional levels[34,35,56]), it is not surprising that we found that modulation of the level of protein in the maternal diet regulates mTORC1 activity resulting in subtle, but distinct changes in the craniofacial shape of the embryos. Availability of nutrition is a major factor in connection with natural selection and such a spectrum of closely related craniofacial shapes may reflect adaptive phenotypic plasticity. Phenotypic plasticity in the feeding apparatus of teleost fish has been observed, both in the wild[57] and under experimental conditions[4]. Recently, it has been reported that the HH pathway mediates plasticity of the feeding apparatus in response to the mechanical properties of the foraging species[27], with mechanical sensing being, at least in theory, mediated by cilia, a

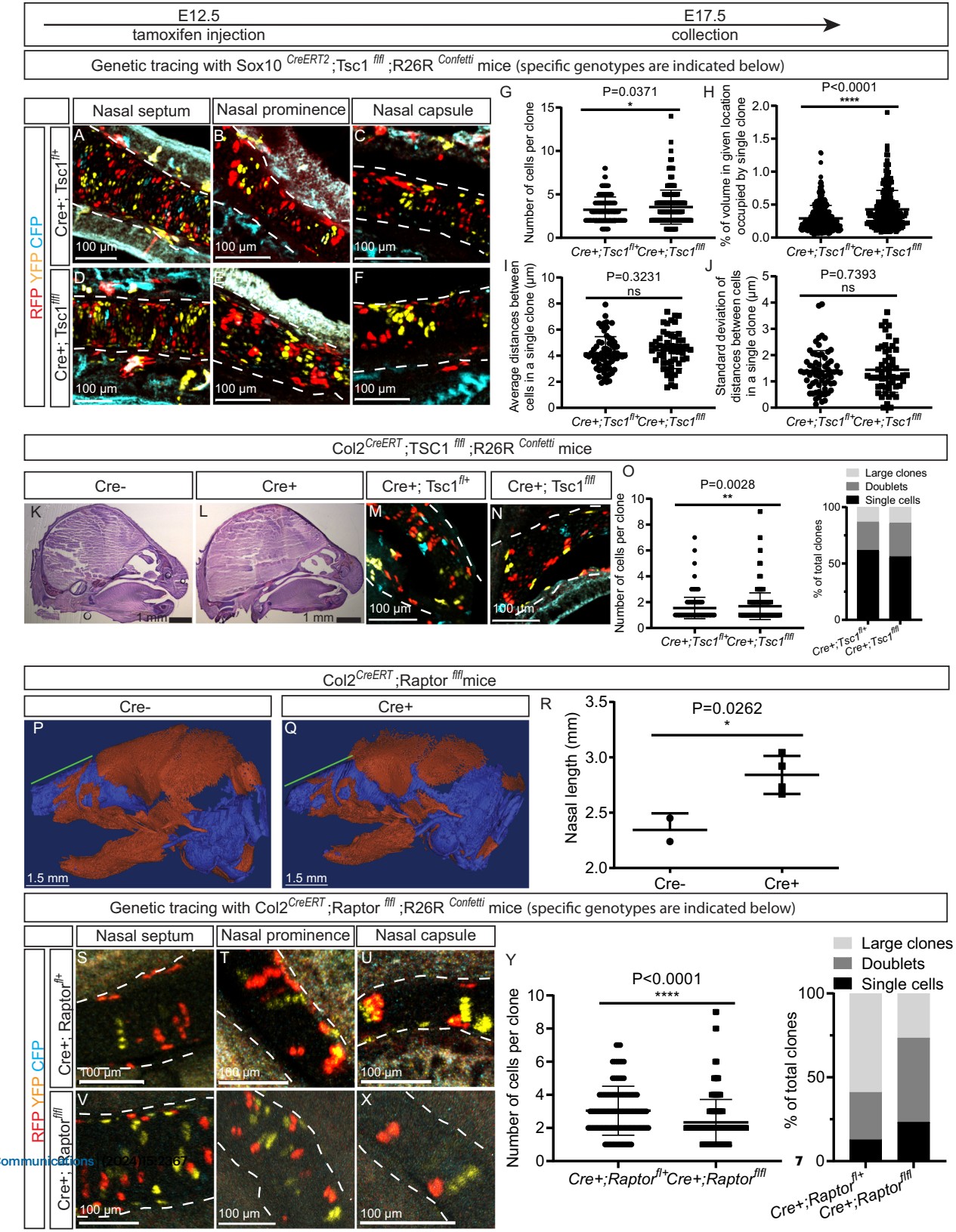

mechanical sensor that is a key component of the HH pathway[26,58,59]. Thus, interactions between nutritional sensing by the mTORC1 pathway and mechanical sensing by the HH pathway may mediate the phenotypic plasticity of the feeding apparatus in response to external conditions. Interestingly, mTORC1 is also involved in regulating the phenotypic plasticity of skeletal muscles[60], as well as in long-term

synaptic plasticity[61]. Thus, the observed mechanism of adaptive phenotypic plasticity may facilitate the abundant production of new phenotypes that may permit some of the offspring to keep up with possible changes in the environment. For example, the differential ability to consume certain prey types can promote trophic divergence and the availability of alternate resources[62].

**Fig. 3 | Modulating mTORC1 activity at the stage of intercalation alters craniofacial shape and the size of chondrocyte clones to a minor extent.** All mice were pulsed with tamoxifen on E12.5 and analyzed on E17.5. **A–J** The mTORC1 pathway was activated by ablation of Tsc1 in chondroprogenitors employing *Sox10$^{CreERT2}$;Tsc1$^{fl/fl}$;R26R$^{Confetti}$* mice. The images shown are representative for the three major types of nasal cartilage - the septum **A, D**, prominence **B, E** and capsule **C, F**. Clones within cartilage were characterized with respect to the number of cells per clone **G**, volume occupied by each clone **H**, average distance between cells within each clone **I**, and dispersion of the clones as reflected in the standard deviation of distances between the cells in each clone **J**. *n* = 3 animals for **G–J**. **K–O** The mTORC1 pathway was activated in chondroprogenitors by ablation of *Tsc1* employing Collagen type 2-driven CreERT recombinase (*Col2$^{CreERT}$;Tsc1$^{fl/fl}$; R26R$^{Confetti}$*). Representative sagittal sections stained with HE **K, L**, clonal appearance **M, N** and quantification of clonal size **O** revealed little difference between the heads

of control (*Cre + ;Tsc1$^{fl+}$;R26R$^{Confetti}$*) **K, M** and *Tsc1* cKO (*Cre + ;Tsc1$^{fl/fl}$;R26R$^{Confetti}$*) **L, N** embryos. *n* = 2 animals in **O**. **P–R** The mTORC1 pathway was inactivated in chondroprogenitors by ablation of the Raptor gene employing Collagen type 2-driven CreERT recombinase (*Col2$^{CreERT}$;Raptor$^{fl/fl}$*). Reconstruction of cartilaginous and bony structures in the heads of (**P**) control (Cre-negative, *Raptor$^{fl/fl}$*) and **Q** *Raptor* cKO (*Col2$^{CreERT}$;Raptor$^{fl/fl}$*) mice was examined by μCT enhanced with Hexabrix. The green line indicating nasal length was quantified **R**. *n* = 2 and 4 animals in **R**. **S–Y** Clonal reporter R26R$^{Confetti}$ was bred into *Col2$^{CreERT}$;Raptor$^{fl/fl}$* mice (*Col2$^{CreERT}$;Raptor$^{fl/fl}$;R26R$^{Confetti}$*) and clonal appearance assessed in the nasal septum **S, V**, nasal prominence **T, W** and nasal capsule **U, X**. The number of cells per clone was quantified **Y**. 118 and 336 clones were quantified in **Y**. Means ± SD are presented for all quantification graphs, with individual values also indicated and determination of statistical significance using the two-sided unpaired t-test. Source data are provided as a Source Data file.

In connection with normal facial variation in humans, several single nucleotide polymorphisms (SNPs) of PI3K/AKT/mTORC1 were identified using GWAS[22]. However, every individual SNP contributes little to the facial appearance. For example, homozygous carriers of the C allele (chr11:17,227,762 C > T) of PIK3C2A gene have a slightly wider nose (approximately 0.5 mm), another SNP of the same gene (chr11:17,124,747 T > C) results in approximately 3 mm longer nasal bridge in homozygous carriers[22]. The SNP of the RRAGA gene (chr9:19,049,512 T > G) causes approximately 2 mm longer middle facial depth in homozygous carriers[22]. At the same time, loss-of-function mutations in either Tsc1 or Tsc2 genes, which are direct intracellular inhibitors of mTORC1, cause a genetic disorder named Tuberous Sclerosis Complex (TSC), characterized by the growth of benign tumors in multiple areas. Pharmacological inhibition of mTORC1 in these patients has demonstrated promising outcomes[63]. Frontal bone thickening and hemi-mandibular expansion have been reported in TSC patients[64]. All these observations indicate the participation of the mTORC1 pathway in craniofacial morphogenesis in humans. Furthermore, humans subjected to a low protein diet decrease their levels of circulating serum IGF1[65], the cytokine that directly signals via the PI3K/AKT/mTORC1 pathway. Thus, it is plausible that the PI3K/AKT/mTORC1 pathway is involved in human craniofacial shaping and may be modulated by the nutritional status. Furthermore, in humans, craniofacial plasticity has been described in response to the consistency of the diet and alcohol consumption by the mother during pregnancy, as well as to climate change[12,66,67]. Thus, the plasticity of the feeding apparatus, as well as of the entire facial skeleton, may be an evolutionarily conserved characteristic of all gnathostomes, from zebrafish to humans. On the basis of the findings of others and the data documented here, we propose that the mTORC1 pathway is a key part of the molecular machinery that adapts craniofacial structures to nutritional conditions.

In summary, we have demonstrated here that the mTORC1 pathway modulates the embryonic shaping of craniofacial skeletal elements at the stage of chondrogenic condensations, with subsequent fine-tuning during intercalation of chondro-progenitors. Furthermore, we provide evidence for an impact of maternal protein intake during pregnancy on the shaping of fetal craniofacial cartilage. These findings provide important insights into the mechanisms underlying craniofacial shaping and, potentially, the phenotypic plasticity of this process as well and, in addition, help elucidate the role of material dietary protein during pregnancy in this context.

## Methods

Here we declare that the analysis of human embryos reported here was authorized by the local ethics committee of the Institute of Fundamental Medicine and Biology of Kazan Federal University, that the study design and conduct complied with all relevant regulations regarding the use of human study participants, and was conducted in accordance with the criteria set by the Declaration of Helsinki.

We further declare that all experiments involving mice and zebrafish reported here were pre-approved by the Stockholm North Ethical Committee and/or Goteborg's Animal Ethical Committee and performed in accordance with the guidelines of the Swedish Animal Agency.

### Human embryos
Human fetal tissue collection was reviewed and approved by the local ethics committee of the Institute of Fundamental Medicine and Biology of Kazan Federal University (No. 8, May 2018), based on the research statement description, stating the study of the contribution of genetic factors to the development of the human face. All the samples used in the study were non-pathologic, free-will-based abortion-derived. Written informed consent was obtained from the patients subjected to medical abortion. The donors were aware of the research purposes before signing the consent. The samples were anonymous, and no donors' personal data were provided to researchers. No specific financial compensation was provided to the donors. A total of 16 embryos, with Post-Conception-Week (PCW) ranging from 3 to 12 were used for analysis. CAGE-seq libraries were created from RNA obtained from the micro-dissected facial area of an individual embryo as specified in a source data file.

To identify enhancers actually transcribed in human embryonic faces, human facial material was collected between weeks 3 and 12 of development, time-window that potentially influence human facial individuality. Next, we performed CAGE-sequencing on embryonic human facial material and compared the transcriptional start sites, proximal promoters and distal transcribed enhancers[68] thus identifying loci indicated as being involved in human facial variability by genome-wide sequencing[22] (http://portaldev.sph.umich.edu/docs/api/v1/#introduction).

For our CAGE-sequencing, total RNA (2–3 μg) was extracted from the facial portion of human embryos, preserved in RNAlate and stored at −80 °C using the RNeasy Fibrous Tissue Kit (Qiagen, Hilden, Germany) in accordance with the manufacturer's protocol. The concentration and purity of extracted RNA were determined on the basis of absorption employing the NanoDrop™ 8000 Spectrophotometer (ThermoFisher Scientific, Waltham, MA, USA) and quality verified with the Agilent Bioanalyzer 2100 (Agilent Technologies, Santa Clara, CA, USA).

Libraries were then prepared utilizing the standard nAnT-iCAGE (non-Amplified non-Tagging Illumina Cap Analysis of Gene Expression) protocol[69], employing 2.5–3 μg total RNA as a template for synthesis of the first cDNA strand (nAnT-iCAGE Library Preparation kit DNA form, Yokohama, Japan and SuperScript III Reverse Transcriptase, Invitrogen, Waltham, MA, USA). This cDNA was subsequently biotinylated at its 5′-end (nAnT-iCAGE Library Preparation kit, DNA form, Yokohama, Japan), which allowed selection of the 5′-cap containing molecules with streptavidin beads (Dynabeads M-270 Streptavidin, ThermoFisher Scientific, USA). In this manner, rRNA, as well as truncated or not fully transcribed RNA was eliminated.

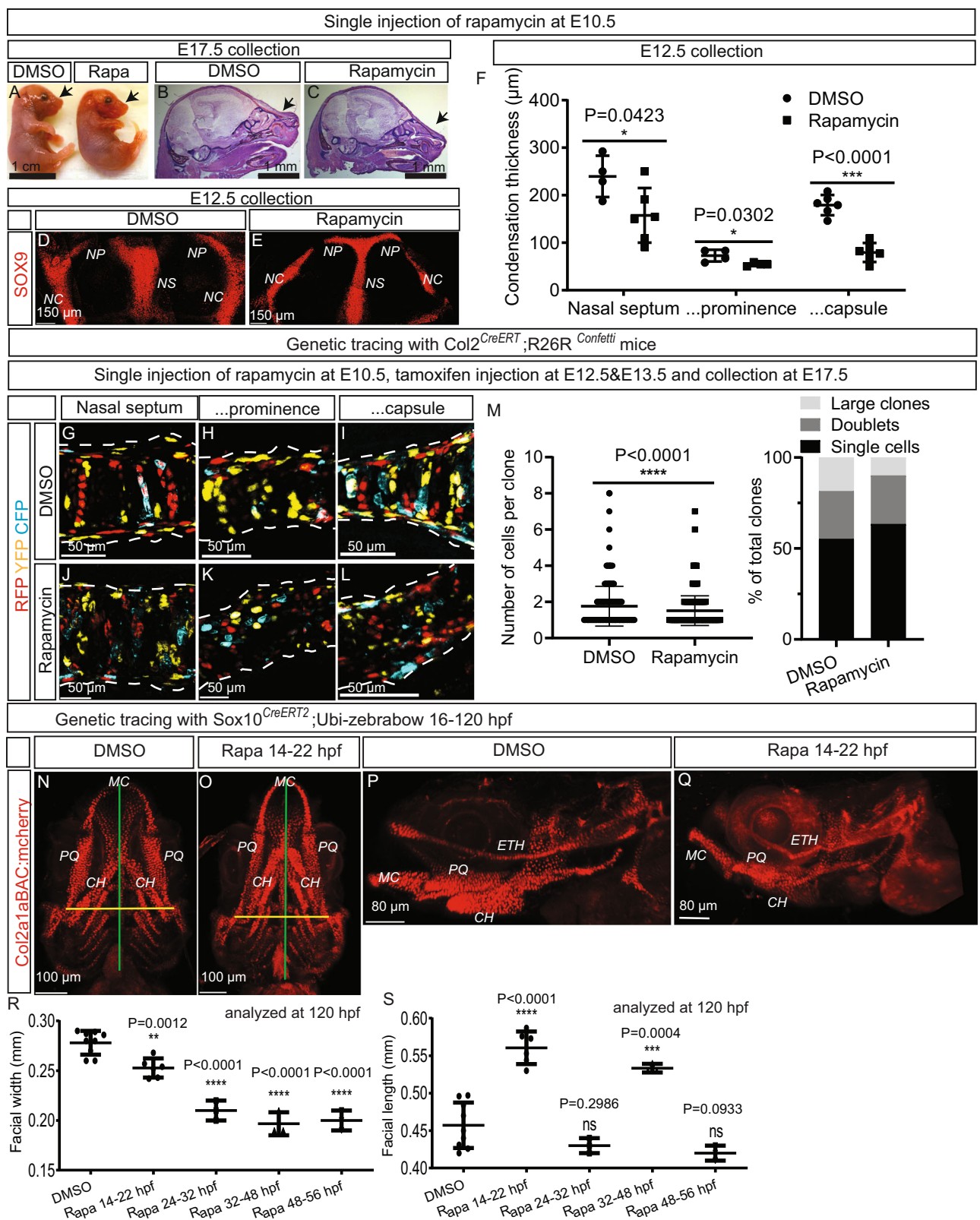

For more complete removal of RNA, the cDNA was treated with RNase I and H (nAnT-iCAGE Library Preparation kit, DNA form, Japan) and then purified using RNACleanUP (Beckman Coulter, Brea, CA, USA). Next, linkers were ligated to the 5′and 3′ ends (nAnT-iCAGE Library Preparation kit, DNAform, Japan) of the cap-trapped cDNA. The 5′- linker employed contained recognition sites for the XmaJI restriction endonuclease (nAnT-iCAGE Library Preparation kit, DNA form, Japan) and the MmeI class II restriction enzyme (nAnT-iCAGE Library Preparation kit, DNA form, Japan), as well as a barcode for multiplexing. The 3′- linker contained a recognition site an XbaI restriction enzyme (nAnT-iCAGE Library Preparation kit, DNAform, Japan). Treatment with these restriction enzymes yielded short CAGE tags to which a sequencing primer was ligated.

**Fig. 4 | Inhibition of mTORC1 at various stages of embryonic development modulates the shape of craniofacial structures in both mice and zebrafish.** **A**–**C** Pregnant C57BL/6 J dams were exposed to a single dose of either DMSO or rapamycin on E10.5 and stained with hematoxylin-eosin on E17.5. Representative images of the general appearance **A** or sagittal sections of their heads **B**, **C** are shown. **D**–**F** Embryos were treated as in A, but stained for SOX9 on E12.5 to reveal chondrogenic condensations **D**, **E**, the thickness of which in the three major nasal compartments was quantified in **F**. *n* = 4 or 6 animals in **F**. **G**–**M** The same treatment as in A was applied to *Col2^CreERT^;R26R^Confetti^* mice pulsed with tamoxifen on E12.5 and E13.5 and clonal appearance analyzed on E17.5 **G**–**L** and clonal size quantified **M**. 685

and 1133 clones obtained from 3 animals were quantified in **M**. **N**–**S** *Col2a1aBAC:mcherry* zebrafish larvae were exposed to DMSO or rapamycin at various periods in development. Ventral **N**, **O** and lateral **P**, **Q** views of the larvae exposed to DMSO **N**, **P** or rapamycin **O**, **Q** 14–22 h post-fertilization (hpf) and imaged at 120 hpf are shown. The facial length **R** and width **P** of 120-hpf-old *Col2a1aBAC:mcherry* zebrafish larvae exposed to rapamycin during the intervals of time indicated were quantified. *n* = 3–9 animals in **R**, **S**. Means ± SD are presented, with individual values also indicated. The two-sided unpaired t-test was employed to compare the values in **F**, **M**, and one-way ANOVA followed by Dunnett's multiple comparisons test in R and S. Source data are provided as a Source Data file.

In the final stage, a second cDNA strand was synthesized from these short CAGE tags (nAnT-iCAGE Library Preparation kit, DNAform, Japan). The concentration of the resulting libraries was determined by the PicoGreen Assay in a GloMax® Multi Detection System (Promega, Madison, WI, USA) and their quality assessed using an Agilent Bioanalyzer 2100 (Agilent Technologies, Santa Clara, CA, USA). Finally, the libraries were validated using real-time PCR (KAPA Library Quantification Kits Illumina, KAPA Biosystems, Wilmington, MA, South Africa) and sequencing on a HiSeq 2500 platform (Illumina, San Diego, CA, USA) using the HiSeq v4 reagent kit (HiSeq SR Cluster Kit v4 cBot and HiSeq SBS Kit v4 50 cycles, Illumina, San Diego, CA, USA) in the 50-bp single-end mode.

Single-read sequences were analyzed for quality and over-represented adapter sequences identified with the FastQC tool. Quality filtering trimming was performed with the fastx_trimmer (FASTX Toolkit 0.0.13.2) and Trimmomatic-0.39 and RNAdust 1.06 utilized as adapters and for removal of rRNA removal. Read mapping on human genome hg38 and mouse genome mm10 was performed with BWA-0.7.10, with unmapped reads being realigned using Hisat2-2.2.1. Aggregation of CAGE tag start sites (CTSS) for each sample, with subsequent peak clustering, were carried out employing the PromoterPipeline script from the C1 CAGE protocol[68]. Bidirectional enhancers were identified using the pipeline described by Andersson and colleagues (2014)[70]. The statistical significance of the differential expression of CAGE peaks was calculated using the edgeR package for R.

Triple overlap of GWAS-derived data of coordinates of face shape-affecting loci with already pre-identified and annotated enhancers (from genome annotation[42]; https://genome.ucsc.edu/cgi-bin/hgTrackUi?db=mm10&g=encodeCcreCombined; ENCODE Project Consortium) and our human CAGE-seq-derived coordinates of active facial human embryonic enhancers was done involving a specific prior filtration step such as: we selected polymorphisms falling within 5 kilo-base pairs distance from the CDS in any direction for GWAS-identified genes.

## Mice

All animal experiments were pre-approved by the Stockholm North Ethical Committee and/or Goteborg's Animal Ethical Committee and performed in accordance with the guidelines of the Swedish Animal Agency. All animals were housed under a 12:12 hour light/dark cycle with free access to food and water. The Sox10-CreERT2, Col2-CreERT2, R26Confetti, Tsc1 flox, and Raptor flox strains of mice employed have been described in detail previously[43,71–74] and were maintained on predominantly C57Bl6 genetic background. Embryonic Cre recombination was induced by intraperitoneal (i.p.) injection of 50 mg tamoxifen (Sigma) per kg of body weight for lineage tracing experiments and 75 mg/kg for experiments involving gene ablation. The day on which the plug was detected was defined as embryonic day 0.5 (E0.5). Rapamycin (0.02 mg, LC Laboratories) was injected i.p. into each pregnant dam. Embryos of both sexes were used.

## Zebrafish

The Col2a1aBAC:mcherry strain of zebrafish was kindly provided by Prof. Chrissy Hammond (University of Bristol, UK) and has been utilized as described in detail elsewhere[75–77]. Zebrafish larvae (up to 120 h

post-fertilization) were exposed to 400 nM rapamycin at the time-points indicated.

## Manipulation of the maternal murine diet

Pregnant dams received standard mouse chow containing 22% protein until E6.5 and thereafter an isocaloric diet containing 4%, 20%, or 40% protein (TD. 93032, TD. 91352 and TD. 90018 from Envigo) until the day of sacrifice.

## X-ray computed microtomography (μCT) with enhanced contrast achieved with phosphotungstic acid (PTA)

On E17.5, the heads of mouse embryos were placed in a 1% PTA/methanol solution to enhance the contrast of soft structures, as described previously[78]. μCT scans were performed with the GE phoenix v|tome|x L 240 system equipped with a nanofocus X-ray tube (180 kV/15 W maximal power) and high flat panel (dynamic 41|100 with 4000 × 4000 pixels, each 100 × 100 μm in size). Acquisition involved the use of a 0.2-mm aluminum filter to soften the beam; 60 kV and 200 μA; exposure for 600 ms; and averaging of 3 projections to reduce noise. 1800 images were acquired over 360°, requiring a scanning time of one hour per sample. The isotropic voxel size was 6.2 μm in all cases. The tomographic reconstructions were performed in the GE phoenix datos|x 2.0 3D computed tomography software. Segmentation of craniofacial structures was performed manually using a combination of the Avizo (Thermo Fisher Scientific, USA) and VG Studio MAX 3.2 software (Volume Graphics GmbH, Germany), as described elsewhere[78].

## High-resolution microfocus computed tomography using enhanced contrast with Hexabrix (CE-HRμCT) and subsequent image processing and 3D analysis

Following fixation, samples were stored in PBS at 4 °C. Prior to scanning, these samples were transferred to Eppendorf tubes containing 1.5 ml 30% Hexabrix 320 in PBS (Guerbet Nederland B.V.); incubated for two weeks with continuous gentle shaking at 4 °C; and then scanned while still inside the same tubes. Hexabrix 320, a negatively charged ioxaglate, is repelled by the anionic sulfated-glycosaminoglycan (sGAG), resulting in negative staining of cartilage, while still providing good contrast between mineralized tissues and the background.

For acquisition of all images, the NanoTom M (GE Measurement and Control Solutions, Germany) system in combination with a diamond-coated tungsten target was employed with the following conditions: a 0.2-mm aluminum filter to soften the beam; 60 kV and 300 μA; exposure for 500 ms; and averaging of each sample individually and a skip of 0 ('fast scan mode'). 2400 images were acquired over 360°, requiring a scanning time of 20 minutes per sample. In all cases the isotropic voxel size was 5 μm. Reconstruction was performed using the Phoenix datos|x CT software, applying a correction of 5 for beam hardening and a Gaussian filter (radius 3) to reduce noise.

The transaxial, coronal and sagittal cross-sections of each sample were visualized with the DataViewer (Bruker MicroCT, Belgium); while 3D visualization of the cartilage and mineralized tissue and quantification

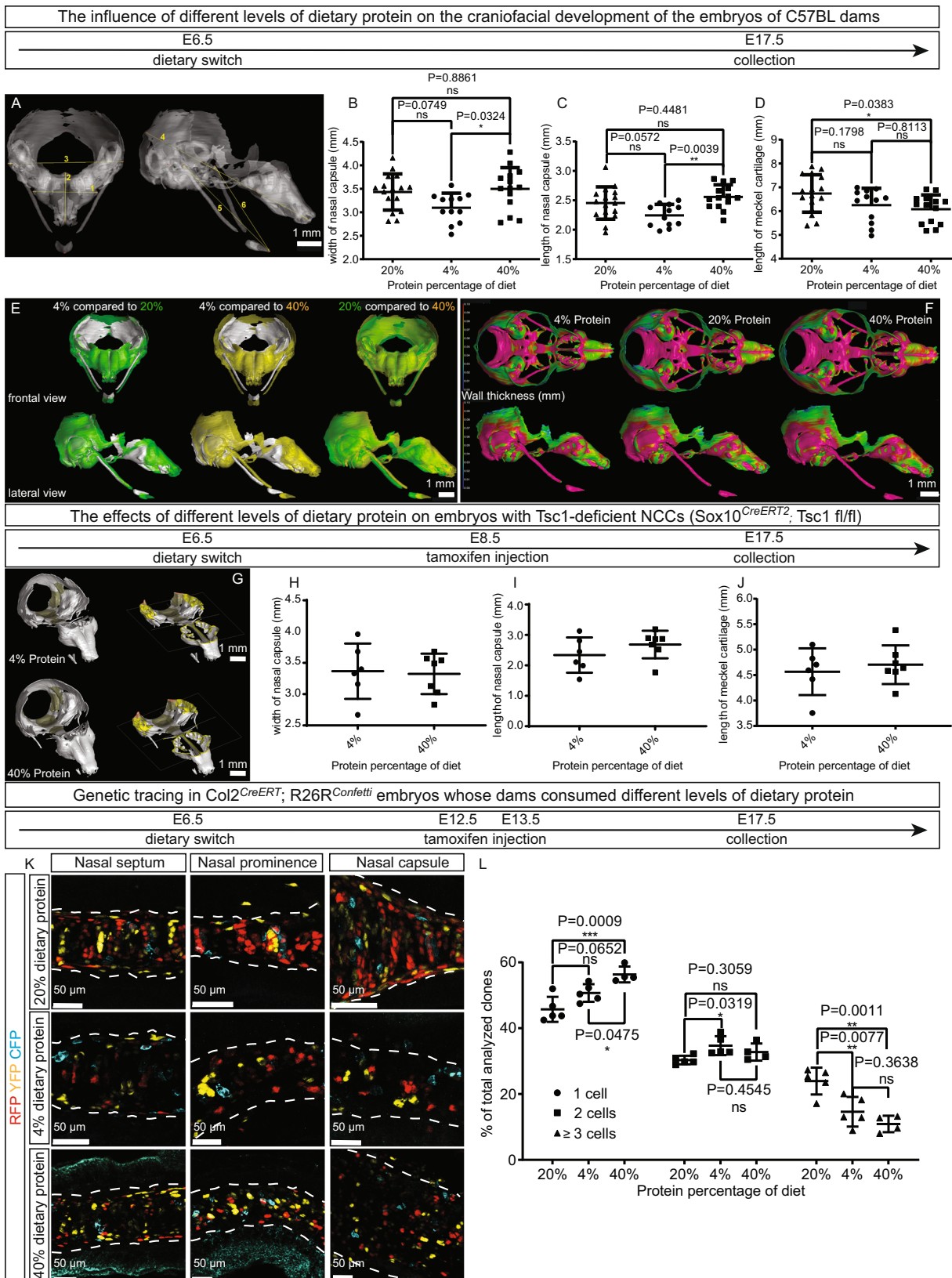

of their volumes were performed with the Mimics Innovation suite (Materialise NV, Belgium). Briefly, two threshold values were selected manually to distinguish between non-mineralized cartilage and mineralized tissues (i.e., mineralized cartilage and subchondral bone) and these thresholds then fine-tuned with dynamic region-growing and multi-slice edit. Using these adjusted threshold values, 3D models based on marching-cubes were generated and the volumes of mineralized tissue versus non-mineralized cartilage and the ratio between these volumes calculated. In addition, the length and width of the nasal capsule and the Meckel cartilage were measured.

**Fig. 5 | The level of protein in the maternal diet alters the structure of the facial skeleton of their embryos. A–F** Genetically identical wild-type pregnant C57BL/6 J dams consumed isocaloric diets containing different levels of protein (4%, 20% (standard chow=22%), 40%) from E6.5 of pregnancy and on E17.5 the skeleton in the heads of their embryos was reconstructed employing μCT with PTA enhancement **A**. The following parameters indicated in A were measured: the thickness (1) and length (2) of the nasal capsule, the thickness (3) and length (4) of the entire chondrocranium, and the length of the right (5) and left (6) portions of the Meckel cartilage. The thickness **B** and length **C** of the nasal capsule and the average length of the Meckel cartilage **D** are shown. Overlayed 3D reconstructions **E** and the thickness of craniofacial structures **F** from embryos whose mothers consumed diets containing different levels of protein are presented for direct comparison. *n* = 12 and 13 animals in **B–D**. **G–J** Pregnant Sox10^{CreERT2};Tsc1fl/fl dams consumed isocaloric diets containing either 4% or 40% protein from E6.5 of pregnancy and Tsc1 was ablated in the neural crest cells of their embryos by pulsing with tamoxifen on E8.5. The skeleton in the heads of their embryos was reconstructed on day E17.5 employing μCT with PTA enhancement **G**. Quantification of the width **H** and length **I** of the nasal capsule and the average length of the Meckel cartilage **J** are depicted. *n* = 6 or 7 animals in **H–J**. **K**, **L** The same diets as in A-F were administered to pregnant *Col2^{CreERT};R26R^{Confetti}* dams pulsed with tamoxifen on E12.5 and E13.5. Representative images of clonal appearance **K** and quantification of clonal size **L** in the entire nasal cartilage are shown. *n* = 4 and 5 animals in **L**. Means ± SD are presented, with individual values also indicated. One-way ANOVA followed by Tukey's multiple comparisons test was employed to compare the values in **B–D** and **L**, and the two-sided unpaired t-test in **H–J**. The white dashed lines in **K** outline the cartilage. Source data are provided as a Source Data file.

## Immunohistochemical analyses
Embryos were fixed in 4% paraformaldehyde (PFA) for 6 hours at 4 °C and tissues then embedded in OCT (Tissue-Tek) on dry ice for sectioning. Thereafter, the 30-μm frozen sections were blocked in PBST (PBS + 0.01% Tween20) + 3% normal horse serum (Vector laboratories) for one hour prior to incubation with the primary antibody (anti-pS6 (Cell Signaling, #2211), anti-SOX9 (Sox9, HPA001758, Sigma Aldrich Inc.), or anti-K67 (Termofisher, MA5-14520)) overnight.

## TUNEL staining
30-μm tissue sections were treated with 10 μg/ml proteinase K (Ambion) for 40 minutes at 37 °C before applying the TUNEL reaction mix (Roche Inc.) for 90 minutes. The cell nuclei were then counterstained with DAPI.

## Staining with haematoxylin and eosin
15-μm frozen sections were stained with haematoxylin for 30 seconds and 0.02% eosin for 2 min.

## Microscopy and image analysis
Images were acquired with an LSM710 confocal microscope. 3D visualization and all quantification were performed utilizing the IMARIS (Bitplane) and ImageJ software.

## Mathematical modeling of human craniofacial anatomy
To investigate the parallels between morphological alterations in mice and their potential manifestations in humans, we conducted a detailed transformation of polygon data representing human craniofacial structures. The original image was extracted from full-body MRI scans (from the BodyParts3D dataset, a resource developed by the Database Center for Life Science in Tokyo, Japan) and transformed as specified below. Our focus was on the human skull, which was dissected into 53 high-resolution segments encompassing an array of teeth, bones, and ligaments.

For the transformation process, we employed the capabilities of Mathematica 11.0, (developed by Wolfram Research in Illinois, USA) using custom-written code (available at https://zenodo.org/records/10363659). Our transformation technique involved a nonlinear 3D transformation algorithm conceptualized as a three-dimensional 'magnifying glass.' This method was adapted from previously published code[79] and featured a radius of 3 cm centered around the nasal cavity.

We calibrated the magnification effect to mirror the expansion observed in the nasal cavity of mice. This allowed us to extrapolate and hypothesize the potential craniofacial changes in humans corresponding to those noted in mouse models. Our algorithm was designed to allow for selective magnification of specific anatomical regions within the skull. This approach ensured that while we magnified certain areas for detailed study, the rest of craniofacial anatomy remained undistorted and true to its original proportions.

## Statistics & reproducibility
A two-tailed unpaired t-test was used when comparing the two groups. When more than two groups were statistically compared, One-Way ANOVA was used with a specified post hoc test to adjust p values for multiple comparisons, and no statistical method was used to predetermine the sample size. All the data were included in the analyses. The experiments were not randomized, and the investigators were not blinded to allocation during experiments and outcome assessment because genotyping was required before analysis. However, clonal analysis was made blindly. Technical replicates were not considered separate observations.

## Reporting summary
Further information on research design is available in the Nature Portfolio Reporting Summary linked to this article.

## Data availability
All the primary data are provided in the Source Data file. Confocal scans are available upon reasonable request. The primary CAGE sequencing data have been deposited at NCBI GEO under accession number GSE254018. Source data are provided with this paper.

## Code availability
The custom-made mathematical code (named anatomical transformations) has been deposited to Zenodo [https://zenodo.org/records/10363659] (https://doi.org/10.5281/zenodo.10363659).

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

## Acknowledgements

We wish to thank Olga Kharchenko for the artwork included, Ostap Dregval for technical assistance and Prof. Seth M. Weinberg (University of Pittsburg) for providing primary data behind GWAS study[22]. This study was supported by the Swedish Research Council (Projects 2020-02298 to A.S.C., 2023-02161 to I.A., 2022-00611 to K.F. and 2021-01805 to M.X.), the Swedish state under the agreement between the Swedish government and the county councils, the ALF-agreement in Gothenburg (ALFGBG-966178 to A.S.C.) and the NovoNordisk Foundation (NNF21OC0070314 to A.S.C.). A.I. acknowledges the financial support from ERC synergy grant "kill-or-differentiate". M.X. was supported by a long-term postdoctoral fellowship from the European Molecular Biology Organization and by Stiftelsen Frimurare Barnhuset i Stockholm. E.I. was supported by a grant from the Russian Basic Science Foundation (#19-29-04115 to A.S.C.). O.G., R.D, G.G., and E.Sh were supported by grant 075-15-2021-1344 by MSHE of the Russian Federation. M.K., T.Z. and J.K. acknowledge financial support in the form of project CEITEC 2020 (LQ1601) from the Ministry of Education, Youth and Sports of the Czech Republic under the National Sustainability Program II and help from the CzechNanoLab Research Infrastructure supported by MEYS CR (LM2018110). M.K. was the recipient of a Ph.D. Talent Scholarship from the Brno City Municipality.

## Author contributions

Conceptualization: M.X. and A.S.C.; Methodology and Investigation: M.X., M.K., Y.G., D.S., O.A., T.Z., G.K., E.I., H.Z., D.B., R.D., G.G., E.S., and P.T.N.; Resources: O.G., K.F., J.K., I.A., and A.S.C.; Writing—Original Draft, Review & Editing: M.X., I.A. and A.S.C; Funding Acquisition: A.S.C., I.A., K.F., M.X., J.K. All authors have read and agreed to submission of the final manuscript for publication.

## Funding

## Competing interests

The authors declare no competing interests.

## Additional information

Meng Xie [1,2,3], Markéta Kaiser [4], Yaakov Gershtein[5], Daniela Schnyder [1,6], Ruslan Deviatiiarov [7,8,9,10], Guzel Gazizova[7], Elena Shagimardanova[7,9], Tomáš Zikmund [4], Greet Kerckhofs[11,12,13,14], Evgeny Ivashkin[15,16], Dominyka Batkovskyte [1], Phillip T. Newton [1,17,18], Olov Andersson [19], Kaj Fried [20], Oleg Gusev[7,8,9,10], Hugo Zeberg [1], Jozef Kaiser [4], Igor Adameyko [1,5] ✉ & Andrei S. Chagin [1,6] ✉

[1]Department of Physiology and Pharmacology, Karolinska Institutet, Stockholm, Sweden. [2]Department of Biosciences and Nutrition, Karolinska Institute, Flemingsberg, Sweden. [3]School of Psychological and Cognitive Sciences, PKU-IDG/McGovern Institute for Brain Research, Peking University, Beijing, China. [4]Central European Institute of Technology, Brno University of Technology, Brno, Czech Republic. [5]Department of Neuroimmunology, Center for Brain Research, Medical University of Vienna, Vienna, Austria. [6]Centre for Bone and Arthritis Research, Institute of Medicine, Sahlgrenska Academy at University of Gothenburg, Gothenburg, Sweden. [7]Regulatory Genomics Research Center, Kazan Federal University, Kazan, Russia. [8]Endocrinology Research Center, Moscow, Russia. [9]Life Improvement by Future Technologies (LIFT) Center, Moscow, Russia. [10]Intractable Disease Research Center, Juntendo University, Tokyo, Japan. [11]Biomechanics Lab, Institute of Mechanics, Materials, and Civil Engineering (iMMC), UCLouvain, Louvain-la-Neuve, Belgium. [12]Pole of Morphology, Institute of Experimental and Clinical Research (IREC), UCLouvain, Woluwe, Belgium. [13]Department of Materials Engineering, KU Leuven, Leuven, Belgium. [14]Prometheus, Division for Skeletal Tissue Engineering, KU Leuven, Leuven, Belgium. [15]A.N. Severtsov Institute of Ecology and Evolution, Russian Academy of Sciences, Moscow, Russia. [16]Department of Developmental and Comparative Physiology, N.K. Koltsov Institute of Developmental Biology, Russian Academy of Sciences, Moscow, Russia. [17]Department of Women's and Children's Health, Karolinska Institutet, Stockholm, Sweden. [18]Astrid Lindgren Children's hospital, Stockholm, Sweden. [19]Department of Cell and Molecular Biology, Karolinska Institutet, Stockholm, Sweden. [20]Department of Neuroscience, Karolinska Institutet, Stockholm, Sweden. ✉e-mail: igor.adameyko@meduniwien.ac.at; andrei.chagin@gu.se

