## [Peer Review File · Nature Communications]

The level of protein in the maternal murine diet modulates the facial appearance of the offspring via mTORC1 signalingREVIEWER COMMENTS

Reviewer #1 (Remarks to the Author):

This is a fascinating manuscript that I feel will be of wide interest to many readers. I regret that my review has been so delayed, if for no other reason that this paper raises so many interesting questions.

Major thoughts:

1. Calling the human genes called out in figure 1 "mTOR" genes is misleading. The only true mTOR gene is RragA; the others regulate other impactors with signals that are transduced to mTORC1. ACTB is probably downstream of the protein-signaling independent mTORC2. The STRING analysis is confusing as it would seem in combination with the preceding human analysis to link PI3K/AKT signaling - not mTOR signaling - strongly to facial development.
2. Why would a low protein diet lower the width of the nasal capsule (from an evolutionary perspective)?
3. Why would both high and low protein have the same effect on the length of meckel cartilage?
4. The data here has clear potential human significance; and the author should discuss if humans with TSC or the Dutch Winter children have mTOR-dependent craniofacial features.
5. The authors might want to cite Solon Biet et al 2014, Cell Metab, who showed that dietary protein level regulates mTORC1 signaling in vivo.

Reviewer #2 (Remarks to the Author):

The manuscript by Xie et al. investigates a potential mechanistic conduit between maternal dietary intake and embryonic regulation of chondrogenic development as a basis for human face shape variation. Integrating analyses of active enhancers in human embryonic facial mesenchyme with previous GWAS interrogation of human face shape variation led the authors to focus on the mTORC1 pathway, which is known to be regulated in part by dietary status. mTORC1 pathway activation via conditional deletion of Tsc1 in cranial neural crest cells resulted in moderate shape changes in certain craniofacial bones and cartilages. Leveraging conceptual and experimental approaches previously established by this group, the authors generate data showing that these changes are, at least in part, attributable to alterations in chondrogenic condensation and subsequent clonal expansion. These findings from genetic manipulation experiments were supported by pharmacological modulation of mTORC1 pathway in mice and zebrafish. To address the central hypothesis, the authors altered protein content of the maternal diet in wildtype mice. Level of maternal dietary protein was found to correlate with mTORC1 pathway activity during embryonic craniofacial development and with shape of certain cartilaginous craniofacial structures. To test whether the dietary impact was specifically modulated through the mTORC1 pathway, a similar assessment was conducted on mice with conditional deletion of Tsc1, with an inconclusive result.

This is a fascinating report in that it brings together multiple conceptual and experimental approaches to address interesting developmental questions with importance to human health and societal interaction. The authors largely leveraged concepts and approaches presented in previous reports from their own and other groups, but their novel integration provides new insight that would be of significant interest to evolutionary and developmental biologists. While the conclusions are generally supported by high quality data, the finding that most directly supports the central thesis (as stated in

the manuscript title) as currently presented is not convincing. This and other major and minor critiques and suggestions are addressed in detail below.

1. The central thesis of the manuscript is that the level of protein in the maternal diet modulates face shape formation through mTORC1 signaling. This premise is generally supported by the finding that protein level correlates with both mTORC1 pathway activity and face shape. The subsequent experiment is designed to determine whether the mTOR pathway is a specific conduit relaying the dietary influence upon development of the craniofacial skeleton. To do this, the authors conduct similar dietary manipulation in *Tsc1* conditional deletion mice. This is a crucial experiment, which unfortunately appears underpowered and lacking in important controls. The data sets from wildtype mice exposed to different maternal diets appear to include ~14 samples in each group and exhibit substantial individual variation, with considerable overlap in the range of values across groups. This degree of variability demonstrates the importance of generating a relatively large sample size for this type of analysis. However, the critical experiments assessing influence of diet in mice with mTORC1 activation appear to evaluate only 3 mice in one dietary condition and 4 in another. While the values appear to situate near those in the highest protein group from the wildtype cohort, they also exhibit substantial variability. Thus the conclusion that there is a lack of any alteration in the craniofacial parameters in *Tsc1* deficient mice is not convincing based upon the evidence provided. Their own data from wildtype mice suggest that the power analysis (used to justify not collecting more samples) was premature. This experiment should also include assessment of control-embryo littermates, which would serve as an important internal positive control for the dietary influence. Increasing the rigor of this experiment is critical because it would provide the most direct evidence to support the assertion stated in the title.

2. The last extended data figure portraying the mathematical extrapolation of shape changes derived from mouse data onto the human craniofacial skeleton is interesting but seems underdeveloped, as there is minimal explanation of how the modeling was performed and no discussion of how the result reflects the natural and well documented face shape variation observed within and across human populations. There seems to be a missed opportunity to contextualize the mouse model findings to the face-shape GWAS used to home-in on mTORC1 signaling at the outset of the manuscript and presented in Figure 1E. Do the mTORC1/dietary-mediated changes observed in mice correspond to human subpopulations with SNPs in mTOR pathway genes?

3. The authors state that embryonic Cre recombination was induced by ip injection of 1-3 mg tamoxifen into each pregnant dam. Does this mean that some dams were given 1 and some 3, or that the dose was normalized to maternal weight? Were controls and conditional knockouts that were compared collected from the same litters and/or exposed to the same dam mass normalized dose of tamoxifen? These are important considerations as tamoxifen administered at similar doses (and in the absence of genetic recombination) has been found to cause embryonic malformations including craniofacial abnormalities (PMIDs 32723574, 34403436).

Minor concerns

1. The authors state that rapamycin injection was used to establish the stage of craniofacial skeletogenesis during which the role of mTORC1 signaling is most important. However, their assessment appears limited to injection at just a single time point in development, which does little to address the stated purpose. Were additional stages of development tested?

2. The authors suggest that their data shown in Figure 1 identified PI3K/mTORC1 signaling as a potentially important determinant in human craniofacial shape. While their analyses of enhancer activity in embryonic human facial mesenchyme and integration with human GWAS data is novel and clever, the statement is an overreach as mTORC1 signaling influencing cranial neural crest biology and craniofacial morphogenesis has been previously reported (PMIDs 29975682, 25639352). The authors later indicate that their findings further confirm these previous reports, which seems a more

appropriate description.

3. The authors state that "mTORC1 activity (as reflected in the level of pS6) was lowest in control embryos and most pronounced in the high group". Presumably, control embryos refers to those in the 20% protein group. However, extended Figure 3 appears to show that the level of pS6 corresponded positively with protein group, such that it was actually lowest in the 4% protein group, highest in the 40% group, and intermediate in the 20% group.

4. In first paragraph of discussion, the authors note that "we have demonstrated that mTORC1 activity in embryos of these species [zebrafish and mice] is modulated by the level of protein in the maternal diet...". It seems that this statement should be restricted to mice.

5. In materials and methods, the authors state that "Pregnant dams received standard mouse chow containing 4% protein until E6.5...". It is stated earlier in the manuscript that typical chow has 22% protein, so this should be clarified.

Please find below our point-to-point response to the reviewers' concerns and suggestions.

REVIEWER COMMENTS

Reviewer #1 (Remarks to the Author):

This is a fascinating manuscript that I feel will be of wide interest to many readers. I regret that my review has been so delayed, if for no other reason that this paper raises so many interesting questions.

We are very grateful to the reviewer for such a high appraisal of our work.

Major thoughts:

1. Calling the human genes called out in figure 1 "mTOR" genes is misleading. The only true mTOR gene is RragA; the others regulate other impactors with signals that are transduced to mTORC1. ACTB is probably downstream of the protein-signaling independent mTORC2. The STRING analysis is confusing as it would seem in combination with the preceding human analysis to link PI3K/AKT signaling - not mTOR signaling - strongly to facial development.

We sincerely apologize for the misleading. It is now more accurately described as PI3K/AKT/mTORC1/autophagy signaling pathway. Specifically, it is now corrected in the abstract (page 1, line 46), introduction (page 3, lines 21-25), the main text (page 3, lines 42-46) the legend to figure 1 as well as in figure 1 itself. We would also like to mention that ACTB is directly regulated by mTORC1 (PMID: 30700035).

We remade the STRING analysis to indicate the connection between identified genes and the PI3K/AKT/mTORC1 signaling pathway. The current results are much more precise and more straightforward, in our opinion (updated Figure 1D).

2. Why would a low protein diet lower the width of the nasal capsule (from an evolutionary perspective)?

We appreciate the reviewer's suggestion to elaborate on this exciting and provocative topic.

We regard the observed changes in the size of facial bones upon nutritional manipulation as a form of individual phenotypic plasticity, which can be defined as the ability of individual genotypes to produce different phenotypes when exposed to different environmental conditions (1); and narrowing it further down to "developmental plasticity", which is defined as the possibility of an organism to modify developmental trajectories in response to specific environmental cues (2). The developmental plasticity of viscerocranium appears more susceptible to epigenetic factors than the neurocranium in mammals (3-5) and modulation of various traits of viscerocranium was reported in response to such diverse stimuli as temperature, nutrition availability, protein content, food hardness, foraging habitat (6-10). Whether this plasticity is underlined by the distinct developmental origin of viscerocranium (i.e., the neural crest is a relatively novel tissue from an evolutionary perspective) is an exciting possibility.

From the perspective of the high developmental plasticity of the entire viscerocranium, we would argue that when considering the potential evolutionary role of this plasticity, it is more accurate to consider

the entire structure than individual traits separately. Thus, it is plausible that variety in the response of individual skeletal elements within the viscerocranium may be, in fact, beneficial, as it facilitates the phenotypic variation of the entire feeding apparatus and, probably, sensory structures. The underlying mechanism (in relation to this manuscript) may include the interaction of the mTORC1 with key factors forming individual skeletal elements within viscerocranium in heterotopic and heterochronic manner, such as BMPs, hedgehogs, Wnts, etc.

Thus, facilitating the abundant production of new phenotypes may permit the offspring to keep up with possible environmental changes. It is not that obvious in mammals but well documented in teleost fish. For example, the differential ability to consume certain prey types can promote trophic divergence and the availability of alternate resources (11).

However, the link between individual phenotypic plasticity and evolution, particularly the inheritance of the valuable phenotypic trait, is more complex. The original experiments supporting the role of phenotypic plasticity in natural selection were generated by Conrad Waddington in the 1950s, who demonstrated that a phenotype initially induced by the environment could be fixed by selection and become constitutively expressed even in the absence of the initial environmental factor, which resulted in so so-called “flexible stem hypothesis” (12). However, genetic assimilation did not happen without genetic variation in the population. At the same time, there are numerous cases of adaptive phenotypic plasticity that cannot be explained by genetic variation. Thus, the role of phenotypic plasticity in natural selection is still controversial and is a matter of debate (13).

While very exciting, we think the evolutionary considerations are slightly beyond the scope of the current manuscript and would appear very speculative in relation to our own observations. At the same time, we included a brief discussion related to phenotypic plasticity (page 8, lines 31-34).

1. Pigliucci, M., Murren, C. J. & Schlichting, C. D. 2006 Phenotypic plasticity and evolution by genetic assimilation. *J. Exp. Biol.* 209, 2362–2367. (doi:10.1242/jeb.02070)
2. Fusco G. Phenotypic plasticity in development and evolution. *Phil. Trans. R. Soc. B* (2010) 365, 547–556 doi:10.1098/rstb.2009.0267
3. Fields, H. W. (1991) Craniofacial growth from infancy through adulthood. *Pediatr. Clin. N. Am.* 38: 1053–1088.
4. Pucciarelli, H. M. (1980) The effects of race, sex, and nutrition on craniofacial differentiation in rats. A multivariate analysis. *Am. J. Phys. Anthropol.* 53: 359–368. ^[L]_[SEP]
5. Pucciarelli, H. M. (1981) Growth of the functional components of the rat skull and its alterations by nutritional effects. *Am. J. Phys. Anthropol.* 56: 33–41
6. Katz DC, Grote MN, Weaver TD (2017) Changes in human skull morphology across the agricultural transition are consistent with softer diets in preindustrial farming groups. *PNAS* 114:9050-9055.
7. Lieberman DE et al., (2004) Effects of food processing on masticatory strain and craniofacial growth in a retrognathic face. *J Hum Evol* 46:655-77
8. Katz DC, Grote MN, Weaver TD (2016) A mixed model for the relationship between climate and human cranial form. *Am J Phys Anthropol* 160:593–603
9. Stewart, T. A. & Albertson, R. C. (2010) Evolution of a unique predatory feeding apparatus: functional anatomy, development and a genetic locus for jaw laterality in Lake Tanganyika scale-eating cichlids. *BMC Biol* 8, 8
10. Klemetsen, A. (2010) The charr problem revisited: exceptional phenotypic plasticity promotes ecological speciation in postglacial lakes. *Freshwater Reviews* 3, 49-74.
11. Tuckett QM, Simon KS, Saros JE, Halliwell DB, Kinnison MT. Fish trophic divergence along a lake productivity gradient revealed by historic patterns of invasion and eutrophication. *Freshw Biol.* 2013;58:2517–31.
12. West-Eberhard MJ (2003) *Developmental plasticity and evolution*. Oxford University Press, New York
13. Massimo Pigliucci. Evolution of phenotypic plasticity: where are we going now?, *Trends in Ecology & Evolution*, Volume 20, Issue 9, 2005, Pages 481-486, <https://doi.org/10.1016/j.tree.2005.06.001>.

3. Why would both high and low protein have the same effect on the length of meckel cartilage?

please see our response to the comment #2 above.

4. The data here has clear potential human significance; and the author should discuss if humans with TSC or the Dutch Winter children have mTOR-dependent craniofacial features.

Thank you for this very useful suggestion. Indeed, craniofacial abnormalities have been reported in patients with mutated TSC-1 and -2 genes. Since Tsc1/2 are direct intracellular mTORC1 inhibitors, it is likely that the observed craniofacial abnormalities are mTOR-dependent. We have elaborated on these observations in the Discussion section (page 8, lines 35-48, page 9, lines 1-3). For Dutch children who were conceived during the famine winter period between 1944 and 1945, poor health status has been observed in their adulthood, including higher risks of cardiovascular disease and higher rates of obesity onset and hospitalization. However, craniofacial structures have not been investigated, and whether mTOR signaling is involved is also unclear to the best of our knowledge.

5. The authors might want to cite Solon Biet et al 2014, Cell Metab, who showed that dietary protein level regulates mTORC1 signaling in vivo.

Thank you very much for bringing this valuable reference to our attention. It is now discussed and cited (page 3, lines 21-25).

Reviewer #2 (Remarks to the Author):

The manuscript by Xie et al. investigates a potential mechanistic conduit between maternal dietary intake and embryonic regulation of chondrogenic development as a basis for human face shape variation. Integrating analyses of active enhancers in human embryonic facial mesenchyme with previous GWAS interrogation of human face shape variation led the authors to focus on the mTORC1 pathway, which is known to be regulated in part by dietary status. mTORC1 pathway activation via conditional deletion of Tsc1 in cranial neural crest cells resulted in moderate shape changes in certain craniofacial bones and cartilages. Leveraging conceptual and experimental approaches previously established by this group, the authors generate data showing that these changes are, at least in part, attributable to alterations in chondrogenic condensation and subsequent clonal expansion. These findings from genetic manipulation experiments were supported by pharmacological modulation of mTORC1 pathway in mice and zebrafish. To address the central hypothesis, the authors altered protein content of the maternal diet in wildtype mice. Level of maternal dietary protein was found to correlate with mTORC1 pathway activity during embryonic craniofacial development and with shape of certain cartilaginous craniofacial structures. To test whether the dietary impact was specifically modulated through the mTORC1 pathway, a similar assessment was conducted on mice with conditional deletion of Tsc1, with an inconclusive result.

This is a fascinating report in that it brings together multiple conceptual and experimental approaches to address interesting developmental questions with importance to human health and societal interaction. The authors largely leveraged concepts and approaches presented in previous

reports from their own and other groups, but their novel integration provides new insight that would be of significant interest to evolutionary and developmental biologists. While the conclusions are generally supported by high quality data, the finding that most directly supports the central thesis (as stated in the manuscript title) as currently presented is not convincing. This and other major and minor critiques and suggestions are addressed in detail below.

We are very pleased that the reviewer found our report fascinating. It is a great pleasure and honor. Below we provide our point-to-point response to the specific concerns raised by the reviewer.

1. The central thesis of the manuscript is that the level of protein in the maternal diet modulates face shape formation through mTORC1 signaling. This premise is generally supported by the finding that protein level correlates with both mTORC1 pathway activity and face shape. The subsequent experiment is designed to determine whether the mTOR pathway is a specific conduit relaying the dietary influence upon development of the craniofacial skeleton. To do this, the authors conduct similar dietary manipulation in *Tsc1* conditional deletion mice. This is a crucial experiment, which unfortunately appears underpowered and lacking in important controls. The data sets from wildtype mice exposed to different maternal diets appear to include ~14 samples in each group and exhibit substantial individual variation, with considerable overlap in the range of values across groups. This degree of variability demonstrates the importance of generating a relatively large sample size for this type of analysis. However, the critical experiments assessing influence of diet in mice with mTORC1 activation appear to evaluate only 3 mice in one dietary condition and 4 in another. While the values appear to situate near those in the highest protein group from the wildtype cohort, they also exhibit substantial variability. Thus the conclusion that there is a lack of any alteration in the craniofacial parameters in *Tsc1* deficient mice is not convincing based upon the evidence provided. Their own data from wildtype mice suggest that the power analysis (used to justify not collecting more samples) was premature. This experiment should also include assessment of control-embryo littermates, which would serve as an important internal positive control for the dietary influence. Increasing the rigor of this experiment is critical because it would provide the most direct evidence to support the assertion stated in the title.

We appreciate this critical suggestion. Indeed, the initial comparison used 3 and 4 *Tsc1*-cKO mice on a 4% and 40% diet, respectively, with subsequent power analysis. We managed to collect 3 more animals per group before our animal house was closed due to parvovirus; the strain was lost during the quarantine. We have analyzed these novel data and combined them with the previous, so we now have 6 and 7 observations per group, respectively. Please see the updated Figure 5h-j.

we have also softened our conclusion (page 6, line 33; page 7, line 2) and incorporated data by others showing that mTORC1 is regulated by protein content in vivo (page 3, lines 21-25). We hope this will satisfy the reviewer.

We would also like to point out that there are several important observations in this manuscript on top of the one proposed by the reviewer above, including the clonal formation of the cartilage, regulation of these clones by mTORC1, regulation of these clones by nutrition, the effect of nutrition on viscerocranial shaping, evolutionary preservation of the observed mechanisms.

2. The last extended data figure portraying the mathematical extrapolation of shape changes derived from mouse data onto the human craniofacial skeleton is interesting but seems underdeveloped, as there is minimal explanation of how the modeling was performed and no discussion of how the result reflects the natural and well documented face shape variation observed within and across

human populations. There seems to be a missed opportunity to contextualize the mouse model findings to the face-shape GWAS used to home-in on mTORC1 signaling at the outset of the manuscript and presented in Figure 1E. Do the mTORC1/dietary-mediated changes observed in mice correspond to human subpopulations with SNPs in mTOR pathway genes?

We apologize for the poor presentation of the modeling. The modeling is now described in more detail (page 12, lines 10-26), and the custom-made code is now deposited in a publicly accessible domain (<https://zenodo.org/records/10363659>).

As suggested, we have explored GWAS data in relation to face shaping. For this, we used data from a genetic association study on human facial morphology (Shaffer et al., 2016) and checked for associations in the RRAGA gene (chr9:19,049,425-19,051,023, hg19) with middle facial depth and for the gene PIK3C2A (chr11:17,108,122-17,229,533, hg19) associations with nasal width and nasal bridge length. These distances have previously been described here (Kesterke et al., 2016).

In the genomic region encompassing RRAGA associated with middle facial depth for the genetic variant rs3177673 (chr9:19,049,512T>G, hg19), homozygous carriers of the T allele have ~2 mm increased middle facial depth. Interestingly, the same variant is associated with facial attractiveness (Hu et al., 2019) for male faces evaluated by female coders ($p = 0.027$, negative association for the T allele). Moreover, a variant in partial linkage disequilibrium (rs4977492, chr9:19,057,551T>C, hg19, $r^2 = 0.13$, (Auton et al., 2015)) is associated with height ($p = 6e-34$ (Yengo et al., 2022)).

For the PIK3C2A gene, we found associations with nasal width for a haplotype tagged by the variant rs214900 (chr11:17,227,762C>T, $p = 1.1e-5$) for which homozygous carriers of the C allele have a slightly wider nose (~0.5 mm). This variant is in strong linkage disequilibrium ($r^2 = 0.95$) with a variant also previously associated with height ($p = 3e-143$ (Yengo et al., 2022)). In addition, we find a set of genetic variants tagged by rs16924900 (chr11:17,124,747T>C) for which homozygous carriers of the C allele have ~3 mm longer nasal bridge ($p = 5.7e-5$).

Thus, the observed changes are on the millimeter scale and just fall in the variation of facial morphology; modeling such minimal changes can be somewhat misleading. Small changes due to known variation do not rule out that the gene products have large effects; it can just be that the naturally occurring genetic variants have minor effects on the mRNA/protein level. Indeed, patients with inactivating mutations of either Tsc1 or Tsc2 show an apparent thickening of the frontal bone and hemi-mandibular expansion.

This is now discussed in detail (page 8, lines 33-48).

References:

- Auton, A., Brooks, L. D., Durbin, R. M., Garrison, E. P., Kang, H. M., Korbel, J. O., Marchini, J. L., McCarthy, S., McVean, G. A., & Abecasis, G. R. (2015). A global reference for human genetic variation. *Nature*, 526(7571), 68–74. <https://doi.org/10.1038/nature15393>
- Hu, B., Shen, N., Li, J. J., Kang, H., Hong, J., Fletcher, J., Greenberg, J., Mailick, M. R., & Lu, Q. (2019). Genome-wide association study reveals sex-specific genetic architecture of facial attractiveness. *PLoS Genetics*, 15(4), e1007973. <https://doi.org/10.1371/journal.pgen.1007973>
- Kesterke, M. J., Raffensperger, Z. D., Heike, C. L., Cunningham, M. L., Hecht, J. T., Kau, C. H., Nidey, N. L., Moreno, L. M., Wehby, G. L., Marazita, M. L., & Weinberg, S. M. (2016). Using the 3D Facial Norms

Database to investigate craniofacial sexual dimorphism in healthy children, adolescents, and adults. *Biology of Sex Differences*, 7, 23. <https://doi.org/10.1186/s13293-016-0076-8>

Shaffer, J. R., Orlova, E., Lee, M. K., Leslie, E. J., Raffensperger, Z. D., Heike, C. L., Cunningham, M. L., Hecht, J. T., Kau, C. H., Nidey, N. L., Moreno, L. M., Wehby, G. L., Murray, J. C., Laurie, C. A., Laurie, C. C., Cole, J., Ferrara, T., Santorico, S., Klein, O., ... Weinberg, S. M. (2016). Genome-Wide Association Study Reveals Multiple Loci Influencing Normal Human Facial Morphology. *PLoS Genetics*, 12(8), e1006149. <https://doi.org/10.1371/journal.pgen.1006149>

Yengo, L., Vedantam, S., Marouli, E., Sidorenko, J., Bartell, E., Sakaue, S., Graff, M., Eliassen, A. U., Jiang, Y., Raghavan, S., Miao, J., Arias, J. D., Graham, S. E., Mukamel, R. E., Spracklen, C. N., Yin, X., Chen, S.-H., Ferreira, T., Highland, H. H., ... Hirschhorn, J. N. (2022). A saturated map of common genetic variants associated with human height. *Nature*, 610(7933), 704–712. <https://doi.org/10.1038/s41586-022-05275-y>

3. The authors state that embryonic Cre recombination was induced by ip injection of 1-3 mg tamoxifen into each pregnant dam. Does this mean that some dams were given 1 and some 3, or that the dose was normalized to maternal weight? Were controls and conditional knockouts that were compared collected from the same litters and/or exposed to the same dam mass normalized dose of tamoxifen? These are important considerations as tamoxifen administered at similar doses (and in the absence of genetic recombination) has been found to cause embryonic malformations including craniofacial abnormalities (PMIDs 32723574, 34403436).

We apologize for the unclear method description. Yes, the dose of tamoxifen was always normalized per body weight. We apologize for the inappropriate description. We have been utilizing 50 ug of Tamoxifen per 1 g of body weight for lineage tracing experiments and 75 ug/g for the experiments where gene ablation was performed. Depending on a mouse's weight and stage of pregnancy, it resulted in 1-3 mg tamoxifen per dam. It is now correctly reflected in the method section (page 10, lines 43-44).

We thank the reviewer for the essential and valuable references. The reference PMID 32723574 indicates that a single injection of even 150 ug/g tamoxifen between E10.5-E11.5 does not cause cleft palate, and 2-3 injections of this high dose are needed to cause malformation. The reference PMID 34403436 utilized 200ug/g dose to cause developmental malformation, and the authors say that "As opposed to 200 mg/kg, a single dose of 50 mg/kg tamoxifen at the same developmental stage did not result in overt structural malformations." This is very much in line with our observations.

Minor concerns

1. The authors state that rapamycin injection was used to establish the stage of craniofacial skeletogenesis during which the role of mTORC1 signaling is most important. However, their assessment appears limited to injection at just a single time point in development, which does little to address the stated purpose. Were additional stages of development tested?

We apologize for the uncertainty in the description. The multiple developmental stages were tested only in zebrafish. The text is now corrected (page 5, lines 12-13).

2. The authors suggest that their data shown in Figure 1 identified PI3K/mTORC1 signaling as a potentially important determinant in human craniofacial shape. While their analyses of enhancer activity in embryonic human facial mesenchyme and integration with human GWAS data is novel and clever, the statement is an overreach as mTORC1 signaling influencing cranial neural crest biology and craniofacial morphogenesis has been previously reported (PMIDs 29975682, 25639352). The authors later indicate that their findings further confirm these previous reports, which seems a more appropriate description.

We thank the reviewer for this critical remark. Indeed, we are well aware of the excellent papers mentioned above. However, both studies are done on mice, and the conclusion we are making based on Figure 1 is entirely related to humans.

3. The authors state that “mTORC1 activity (as reflected in the level of pS6) was lowest in control embryos and most pronounced in the high group”. Presumably, control embryos refers to those in the 20% protein group. However, extended Figure 3 appears to show that the level of pS6 corresponded positively with protein group, such that it was actually lowest in the 4% protein group, highest in the 40% group, and intermediate in the 20% group.

We thank the reviewer for the sharp eye and apologize for the mistake. The lowest levels of pS6 were observed in 4%-protein group, not in control. It is now corrected (page 6, line 12).

4. In first paragraph of discussion, the authors note that “we have demonstrated that mTORC1 activity in embryos of these species [zebrafish and mice] is modulated by the level of protein in the maternal diet....”. It seems that this statement should be restricted to mice.

We apologize for the unclearness. The statement is now restricted to mouse embryos (page 7, line 12). We thank the reviewer for bringing it to our attention.

5. In materials and methods, the authors state that “Pregnant dams received standard mouse chow containing 4% protein until E6.5...”. It is stated earlier in the manuscript that typical chow has 22% protein, so this should be clarified.

This is now corrected (page 11, line 7). We again thank the reviewer for their sharp eye and attention to the detail.

We hope the reviewer will find the revised manuscript of satisfactory quality.

Again, we thank both reviewers for the excellent, very constructive, and helpful suggestions to improve the quality of our work.

REVIEWERS' COMMENTS

Reviewer #1 (Remarks to the Author):

The authors have done an excellent job responding to the numerous critiques, and this revised manuscript is now suitable for publication. It will undoubtedly be of interest to a broad spectrum of readers. I note that the authors cite Solon-Biet 2014 for the proposition that a PR diet restricts mTORC1 signaling, which is true as far as it goes - and it goes only as far as the liver. For the general proposition, the ability of PR to reduce mTORC1 signaling in mammals was hypothesized by James Mitchell (PMID: 23216249), and perhaps by others earlier; and it was shown to be true in multiple tissues in the mouse by Luigi Fontana (PMID: 26378060). Discussing these and any other relevant references would be appropriate.

Reviewer #2 (Remarks to the Author):

The authors have thoughtfully addressed each of the concerns, comments, and corrections provided in my initial review. Addressing these issues has benefited the clarity of the manuscript, the rigor of the data, and the discussion of results. The strengths of the study and importance of the findings as noted in my original review are maintained in the revised version.

REVIEWERS' COMMENTS

Reviewer #1 (Remarks to the Author):

The authors have done an excellent job responding to the numerous critiques, and this revised manuscript is now suitable for publication. It will undoubtedly be of interest to a broad spectrum of readers. I note that the authors cite Solon-Biet 2014 for the proposition that a PR diet restricts mTORC1 signaling, which is true as far as it goes - and it goes only as far as the liver. For the general proposition, the ability of PR to reduce mTORC1 signaling in mammals was hypothesized by James Mitchell (PMID: 23216249), and perhaps by others earlier; and it was shown to be true in multiple tissues in the mouse by Luigi Fontana (PMID: 26378060). Discussing these and any other relevant references would be appropriate.

We are grateful to the reviewer for finding our revision excellent. We also appreciate the reviewer's suggestion regarding the literature cited. Regulation of mTORC1 by protein restriction (PR) is extensively described in cited reviews (references in the R1 version #33, 54, 55, 56 and the current R2 version - #33, 54, 56, 57). But, of course, the original works should be appreciated. Accordingly, as suggested, we have also cited the paper by the Luigi Fontana group (PMID:26378060; reference #55 in the R2 resubmitted manuscript).

33 Liu, G. Y. & Sabatini, D. M. mTOR at the nexus of nutrition, growth, ageing and disease. *Nat Rev Mol Cell Biol* 21, 183-203 (2020). <https://doi.org/10.1038/s41580-019-0199-y>

54. Kim, J. & Guan, K. L. mTOR as a central hub of nutrient signalling and cell growth. *Nat Cell Biol* 21, 63-71 (2019). <https://doi.org/10.1038/s41556-018-0205-1>

55 Bar-Peled, L. & Sabatini, D. M. Regulation of mTORC1 by amino acids. *Trends Cell Biol* 24, 400-406 (2014). <https://doi.org/10.1016/j.tcb.2014.03.003>

56 Sengupta, S., Peterson, T. R. & Sabatini, D. M. Regulation of the mTOR complex 1 pathway by nutrients, growth factors, and stress. *Mol Cell* 40, 310-322 (2010). <https://doi.org/10.1016/j.molcel.2010.09.026>

Reviewer #2 (Remarks to the Author):

The authors have thoughtfully addressed each of the concerns, comments, and corrections provided in my initial review. Addressing these issues has benefited the clarity of the manuscript, the rigor of the data, and the discussion of results. The strengths of the study and importance of the findings as noted in my original review are maintained in the revised version.

We thank the reviewer for the valuable help and suggestions that allowed us to improve the manuscript.